# A Machine-Learning-Algorithm-Based Prediction Model for Psychotic Symptoms in Patients with Depressive Disorder

**DOI:** 10.3390/jpm12081218

**Published:** 2022-07-26

**Authors:** Kiwon Kim, Je il Ryu, Bong Ju Lee, Euihyeon Na, Yu-Tao Xiang, Shigenobu Kanba, Takahiro A. Kato, Mian-Yoon Chong, Shih-Ku Lin, Ajit Avasthi, Sandeep Grover, Roy Abraham Kallivayalil, Pornjira Pariwatcharakul, Kok Yoon Chee, Andi J. Tanra, Chay-Hoon Tan, Kang Sim, Norman Sartorius, Naotaka Shinfuku, Yong Chon Park, Seon-Cheol Park

**Affiliations:** 1Department of Psychiatry, Kangdong Sacred Heart Hospital, Hallym University College of Medicine, Seoul 05355, Korea; kkewni@gmail.com; 2Department of Neurosurgery, Hanyang University College of Medicine, Seoul 05355, Korea; ryujeil@hanyang.ac.kr; 3Department of Neurosurgery, Hanyang University Guri Hospital, Guri 11923, Korea; 4Department of Psychiatry, Inje University Haeundae Paik Hospital, Busan 47392, Korea; bongjulee@empal.com; 5Department of Psychiatry, Presbyterian Medical Center, Jeonju 54987, Korea; irene.h.na@gmail.com; 6Unit of Psychiatry, Department of Public Health and Medicinal Administration, Institute of Translational Medicine, Faculty of Health Sciences, University of Macau, Macao SAR 999078, China; xyutly@gmail.com; 7Department of Neuropsychiatry, Graduate School of Medical Sciences, Kyushu University, Fukuoka 812-8582, Japan; kanba.shigenobu.921@m.kyushu-u.ac.jp (S.K.); kato.takahiro.015@m.kyushu-u.ac.jp (T.A.K.); 8Department of Psychiatry, Kaohsiung Chang Gung Memorial Hospital, Kaohsiung & Chang Gung University School of Medicine, Taoyuan 83301, Taiwan; chongmy@live.com; 9Psychiatry Center, Tapei City Hospital, Taipei 300, Taiwan; sklin@tpech.gov.tw; 10Department of Psychiatry, Post Graduate Institute of Medical Education and Research, Chandigarh 133301, India; drajitavasthi@yahoo.co.in (A.A.); drsandeepg2002@yahoo.com (S.G.); 11Pushpagiri Institute of Medical Sciences, Tiruvalla 689101, India; roykalli@gmail.com; 12Department of Psychiatry, Faculty of Medicine Siriraj Hospital, Mahidol University, Bangkok 10400, Thailand; pornjirap@gmail.com; 13Tunku Abdul Rahman Institute of Neurosciences, Kuala Lumpur 5600, Malaysia; cheekokyoon@yahoo.com; 14Department of Psychiatry, Faculty of Medicine, Hasanuddin University, Makassar 90245, Indonesia; ajtanra@yahoo.com; 15Department of Pharmacology, National University Hospital, Singapore 119074, Singapore; chay_hoon_tan@nuhs.edu.sg; 16Institute of Mental Health, Buangkok Green Medical Park, Singapore 539747, Singapore; ksim6133@gmail.com; 17Association for the Improvement of Mental Health Programmes, 1211 Geneva, Switzerland; artorius@normansartorius.com; 18Department of Social Welfare, School of Human Sciences, Seinan Gakuin University, Fukuoka 814-8511, Japan; shinfukunaotaka@gmail.com; 19Department of Psychiatry, Hanyang University College of Medicine, Seoul 04763, Korea; hypyc@hanyang.ac.kr; 20Department of Psychiatry, Hanyang University Guri Hospital, Guri 11923, Korea

**Keywords:** psychotic symptoms, depressive disorders, major depression, machine learning, precision medicine

## Abstract

Psychotic symptoms are rarely concurrent with the clinical manifestations of depression. Additionally, whether psychotic major depression is a subtype of major depression or a clinical syndrome distinct from non-psychotic major depression remains controversial. Using data from the Research on Asian Psychotropic Prescription Patterns for Antidepressants, we developed a machine-learning-algorithm-based prediction model for concurrent psychotic symptoms in patients with depressive disorders. The advantages of machine learning algorithms include the easy identification of trends and patterns, handling of multi-dimensional and multi-faceted data, and wide application. Among 1171 patients with depressive disorders, those with psychotic symptoms were characterized by significantly higher rates of depressed mood, loss of interest and enjoyment, reduced energy and diminished activity, reduced self-esteem and self-confidence, ideas of guilt and unworthiness, psychomotor agitation or retardation, disturbed sleep, diminished appetite, and greater proportions of moderate and severe degrees of depression compared to patients without psychotic symptoms. The area under the curve was 0.823. The overall accuracy was 0.931 (95% confidence interval: 0.897–0.956). Severe depression (degree of depression) was the most important variable in the prediction model, followed by diminished appetite, subthreshold (degree of depression), ideas or acts of self-harm or suicide, outpatient status, age, psychomotor retardation or agitation, and others. In conclusion, the machine-learning-based model predicted concurrent psychotic symptoms in patients with major depression in connection with the “severity psychosis” hypothesis.

## 1. Introduction

Depressive disorders are an important issue in the field of mental health care [1]. Although psychotic symptoms are infrequently associated with depressive disorders [2,3,4], the clinical condition of depression accompanied by psychotic symptoms is termed “psychotic depression”. The Psychotic Depression Assessment Scale, which consists of six items from the Hamilton Melancholia Subscale and five items from the Brief Psychiatric Rating Scale, is a psychometric assessment tool used to differentiate depression with psychotic features from major depressive disorder and evaluate the severity of psychotic major depression [5,6]. In addition, psychotic major depression has been defined according to the Kraepelinian dichotomous view. The *Diagnostic and Statistical Manual of Mental Disorders*, second edition (DSM-II), defines psychotic depressive reactions as severe depressive episodes in response to one or more identifiable stressors, in which depressive disorders with mood-incongruent delusions are considered a subtype of schizophrenia or schizoaffective disorder. The DSM-III and DSM-IV classify psychotic major depression as a subtype of severe depression or a severe variant of depression, according to the “severity psychosis” hypothesis that severe levels of depression are closely related to psychotic symptoms. However, in the DSM-5, with the rejection of the “severity psychosis” hypothesis, the specifier “with psychotic features” was added in connection with any level of depressive episodes, including dysthymia, mild and moderate depression, and severe depression [7,8,9,10,11,12,13]. Østergaard et al. [8] proposed that psychotic major depression can be defined as a distinctive diagnostic entity of “meta-syndrome,” including both unipolar and bipolar psychotic depression. However, concerning psychiatric taxonomy, it remains debatable whether psychotic major depression is a subtype of major depression or a clinical syndrome distinct from non-psychotic major depression.

Psychotic major depression is distinct from non-psychotic major depression as follows [10,11,12,13]. First, psychotic major depression is symptomatically characterized by a greater severity of the depressive episode, higher levels of psychomotor disturbances and appetite or weight loss, higher rates of rumination and insomnia, and greater deficits in cognitive performance compared to non-psychotic major depression [14,15,16,17,18]. Second, psychotic major depression is clinically characterized by a longer duration of each subsequent episode; increased vulnerability of conversion to bipolar disorder; greater familial prevalence of bipolar disorder and major depressive disorder; and greater rates of recurrence, suicide, and mortality compared to non-psychotic major depression [19,20,21,22,23,24,25,26]. Third, psychotic major depression is neurobiologically characterized by increased activity of the hypothalamic–pituitary–adrenal axis, smaller volume of the higher associative regions of the frontal and insular cortices, and lower activity of dopamine-β-hydroxylase compared to non-psychotic major depression [27,28,29,30,31,32,33,34,35]. Lastly, regarding pharmacological treatment, the combination of an antidepressant with an antipsychotic is recommended for patients with depression with psychotic features [36,37,38,39]. Additionally, a phase 3 trial reported that mifepristone, which is used to terminate early pregnancy by blocking progesterone action and is a potent reversible antagonist of glucocorticoid receptors, showed a rapid therapeutic response in patients with depression and psychotic symptoms [40].

In the context of precision medicine [41], individualized treatment and prevention of long-term complications and sequelae require the accurate detection of psychotic symptoms (i.e., delusions and hallucinations) in the clinical manifestations of depressive disorders. However, to our knowledge, few predictive models have been reported for concurrent psychotic symptoms in depression. Thus, the current study developed a machine-learning-algorithm-based prediction model for concurrent psychotic symptoms in patients with depressive disorders using data from the Research on Asian Psychotropic Prescription Patterns for Antidepressants (REAP-AD) survey, the largest international research collaboration on psychiatry in Asian regions [41,42,43].

## 2. Materials and Methods

### 2.1. Study Overviews and Participants

The REAP-AD enrolled 2320 consecutive psychiatric patients who used antidepressants from March to June 2013. The patients were enrolled using a convenience sampling method from 39 research centers in 10 Asian countries and special administrative regions (SARs): namely, mainland China, Hong Kong, India, Indonesia, Japan, Korea, Malaysia, Singapore, Taiwan, and Thailand. This study has been described in detail elsewhere [42,43,44]. In addition, using the United Nations classification, the 10 Asian countries and SARs were geographically divided into eastern Asia (mainland China, Hong Kong, Japan, Korea, and Taiwan), southern Asia (India), and southeast Asia (Indonesia, Malaysia, Singapore, and Thailand). Furthermore, using the World Bank list of economies, the countries and SARs were economically categorized as high-income (Hong Kong, Japan, Korea, Singapore, and Taiwan), upper-middle (China, Malaysia, and Thailand), and lower-middle income (India and Indonesia) and SARs [42,43,44]. The REAP-AD was approved by the Institutional Review Board of Taipei City Hospital, Taipei, Taiwan (receipt number: TCHIRB-1020206-E), and other participating centers. All participants provided written informed consent.

The current study used only data from participants meeting the following inclusion criteria: (i) confirmed psychiatric diagnosis of a depressive episode (F32) or recurrent depressive disorder (F33), under the 10th revision of the International Statistical Classification of Diseases and Related Health Problems (ICD-10) Classification of Mental and Behavioral Disorders [45]; (ii) medication with antidepressants, coded as N06A under the Anatomical Therapeutic Chemical classification system [46]; (iii) age >18 and <80 years; and (iv) availability of data on the presence or absence of psychotic symptoms. We excluded data from the following participants: (i) psychiatric comorbidity of organic mental disorders, schizophrenia, bipolar disorders, or intellectual disorder under the ICD-10 Classification of Mental and Behavioral Disorders [45], and (ii) physical comorbidity of seizure disorders, other neurological disorders, or severe physical diseases. Under the ICD-10 Classification of Mental and Behavioral Disorders [45], the specifier “with psychotic symptoms” can be coded only in severe depression but not in mild and moderate depression. However, as mentioned earlier, psychotic symptoms can be present in dysthymia, mild to moderate depression, and severe depression. The clinical condition of depression accompanied by concurrent psychotic symptoms is denoted as “psychotic depression”. Thus, irrespective of the presence or absence of concurrent psychotic symptoms, the ICD-10 diagnostic entities were coded exclusively for the severity of depressive episodes or depressive disorder. To increase the validity and reliability of the ICD-10 diagnoses, consensus meetings were held before the initiation of the REAP-AD survey. Finally, 1171 Asian patients with depressive episodes or recurrent depressive disorders were included in the current analysis.

### 2.2. Demographic Characteristics, Psychotic Symptoms, and Depressive Symptom Profiles

Age and sex were collected as demographic characteristics. The birth season was defined as described by Fountoulakis et al. [47] and Pjrek et al. [48] as follows: spring, March to May; summer, June to August; autumn, September to November; and winter, December to February. Concurrent psychotic symptoms (i.e., delusions and hallucinations) were defined based on the ICD-10 Classification of Mental and Behavioral Disorders [45]. The 10 depressive symptom profiles (i.e., depressed mood, loss of interest and enjoyment, reduced energy and diminished activity, reduced concentration and attention, reduced self-esteem and self-confidence, ideas of guilt and unworthiness, psychomotor agitation or retardation, ideas or acts of self-harm or suicide, disturbed sleep, diminished appetite) were defined based on the National Institute for Health and Care Excellence (NICE) guidelines for the Treatment and Management of Depression in Adults [49]. In addition, the degree of depression was operationally defined by the total number of depression symptoms and categorized into four groups based on NICE guidelines [49]: subthreshold (<4 symptoms), mild (4 symptoms), moderate (5–6 symptoms), and severe (>6 symptoms). Moreover, comorbid psychiatric symptoms (i.e., anxiety symptoms and somatic symptoms) and psychiatric comorbidity (i.e., mental and behavioral disorders due to psychoactive substance abuse [F1], neurotic, stress-related, and somatoform disorder [F4]) were defined based on the ICD-10 [45]. Admissions as outpatients or inpatients were also determined.

### 2.3. Variable Profiles for the Prediction Model of Concurrent Psychotic Symptoms

This study used sociodemographic data (i.e., region (eastern Asia, southern Asia, or southeastern Asia), income level (i.e., high, upper-middle, or lower-middle income level countries), age, sex, or season of birth (i.e., spring, summer, autumn, or winter)), the ten individual depressive symptom profiles, degree of depression (i.e., subthreshold, mild, moderate, or severe), comorbid anxiety or somatic symptoms, psychiatric comorbidity (i.e., substance use disorder [F1] or anxiety and somatoform disorder [F4]), and outpatient or inpatient admission as variables for the prediction model of concurrent psychotic symptoms.

### 2.4. Data Processing and Machine Learning

SAS Enterprise Guide 7.1 (SAS Institute Inc., Cary, NC, USA) was used to extract the necessary data from the REAP-AD dataset. R 3.3.0 [50] and RStudio 1.0.136 [51] were used for all subsequent analyses. The dataset was divided into training and test sets at a 7:3 ratio among the 1171 patients with depressive disorders. The synthetic minority oversampling technique [52] was used due to the low prevalence of psychotic symptoms in patients with depressive disorders. The data were not structurally transformed into the test set. Ten-fold cross-validation was used to train the entire training dataset. The area under the curve (AUC) of the receiver operating characteristic (ROC) was used to establish the performance of the prediction model for concurrent psychotic symptoms in patients with depressive disorder. In addition, the overall accuracy, sensitivity, specificity, negative predictive value, and positive predictive value (PPV) were calculated. The caret library was used to evaluate the performance indicators (i.e., hyperparameter tuning, confusion matrix composition, and AUC).

Although the decision tree model can provide an intuitive sense of variable importance and classification mechanisms with low computing cost, it is vulnerable to overfitting. Therefore, this study used a random forest because its predictive power is maximized when new data (i.e., test set) are provided rather than used for training by minimizing the overfitting of the decision tree [53]. A random forest is a collection of decision trees. A collection of 500 decision trees was generated and integrated in a random forest in this study. Given this advantage, practical problem-solving algorithms in various fields typically use random forests [54,55]. The principle of “mean decreases in accuracy” was used to calculate the variable importance to identify which variables contributed significantly to the optimization of the predictive model [56]. The importance of variable *j* was calculated according to the following process. Several cases were not sampled, which were termed “out-of-bag” because they were permitted to be sampled more than once for the same classifier by bagging.

The training performance of utilizing one of the trees contributes to the handing down of the out-of-sample to the tree and decreases the prediction accuracy. Along with the subsequent change in variable *j* among the out-of-bag samples at random, the accuracy was recalculated. Before and after permutation over all trees, averaging the gap in out-of-bag errors was used to calculate the raw score for the importance of variable *j*. Subsequently, the standardized deviation of the difference was used to return the score to normal. Finally, the score was reduced such that the minimum and maximum values were set to 0 and 100, respectively. The prediction performance of the out-of-bag portion of the data was recorded for each tree. The prediction performance was measured and recorded during each alternative permutation of the predictors. Based on all trees, the average value of the difference between the two prediction performances was obtained and normalized according to the standard error. The test data, which were not used in the training phase, were used to measure the final prediction performance.

## 3. Results

### 3.1. Participant Characteristics

A total of 1171 Asian patients with depressive disorders (240, 38, 142, 173, 38, 50, 130, 144, 144, and 107 Chinese, Hong Kong, Japanese, Korean, Singaporean, Taiwanese, Indian, Malaysian, Thai, and Indonesian patients, respectively) were evaluated (Table 1). Only 1.4% (*n* = 16) of the study participants presented with concurrent psychotic symptoms including delusions and/or hallucinations. Compared to those without psychotic symptoms, patients with depressive disorder psychotic symptoms were characterized by significantly greater rates of depressed mood (χ^2^ = 5.969, *p* = 0.015), loss of interest and enjoyment (χ^2^ = 5.216, *p* = 0.022), reduced energy and diminished activity (χ^2^ = 5.617, *p* = 0.018), reduced self-esteem and self-confidence (χ^2^ = 4.001, *p* = 0.045), ideas of guilt and unworthiness (χ^2^ = 9.527, *p* = 0.003), psychomotor agitation or retardation (χ^2^ = 6.879, *p* = 0.009), disturbed sleep (χ^2^ = 6.303, *p* = 0.012), diminished appetite (χ^2^ = 9.575, *p* = 0.002), and greater proportions of moderate and severe degrees of depression (χ^2^ = 21.104, *p* < 0.0001).

### 3.2. Prediction Model Performance for Concurrent Psychotic Symptoms and Its Variable Importance

A collection of 500 decision trees, which were generated in this analysis, was used by a random forest. An example of 500 decision trees was presented, as shown in Figure 1. The AUC under the ROC curve was 0.823 (Figure 2). The overall accuracy was 0.931 (95% confidence interval, 0.897–0.956), the sensitivity was 0.047, the specificity was 0.993, the positive predictive value was 0.333, and the negative predictive value was 0.937. Severe levels of depression (degree of depression) were the most important variables in the prediction model for concurrent psychotic symptoms, followed by diminished appetite, subthreshold (degree of depression), ideas or acts of self-harm or suicide, outpatient status, age, psychomotor retardation or agitation, and others (Figure 3).

## 4. Discussion

In this study, severe depression (degree of depression) was the most important variable in the prediction model, showing the highest statistical coefficient (χ^2^ = 21.104) among the prediction model variables. This finding was also consistent with the “severity psychosis” hypothesis [7,8,9,10,11,12,13]. In the ICD-10 Classification of Mental and Behavioral Disorders [45], the specifier “with psychotic symptoms” is permitted only for severe depression but not for mild depression and moderate depression, based on the “severity psychosis” hypothesis. However, Østergaard et al. [9] reported that depression was weakly correlated with hallucinations and delusions in 357 patients with severe depression. Thus, psychotic depression may be a clinical syndrome distinct from non-psychotic depression. As such, this finding may be attributed to the possibility that concurrent psychotic symptoms are dependent on the severity of depression or that the specifier “with psychotic symptoms” is coded only for severe levels of depression in the operational definition of the ICD-10 [45]. Moreover, the specifier “with psychotic symptoms” will be coded not only for severe depressive episodes but also for moderate depressive episodes in the ICD-11 Coding Tool [57]. Further studies based on the diagnostic system independent of the “severity psychosis” hypothesis may be required to clarify concurrent psychotic symptoms in patients with depressive disorders. In addition, subthreshold depression and outpatient admission were the third and fourth most important variables in the prediction model, respectively. These findings are also related to the “severity psychosis” hypothesis.

Depressive symptom profiles, including diminished appetite, ideas or acts of self-harm or suicide, psychomotor agitation or retardation, and reduced self-esteem and self-confidence, were important variables in the prediction model. In addition to psychotic symptoms, other clinical characteristics can also be hallmarks differentiating psychotic from non-psychotic major depression [9]. Our findings are consistent with the significantly higher level of psychomotor disturbances and appetite or weight loss and greater rates of suicide and mortality in patients with psychotic major depression compared to those without [10,11,12,13]. However, although depressed mood, reduced energy, diminished activity, and loss of interest and enjoyment are the core symptoms of depressive episodes in the ICD1-10 [45], it is remarkable that these traits were relatively less important variables than other depressive symptom profiles in the prediction model. Thus, these findings cannot be explained easily. However, they can partly support psychotic major depression as a clinical syndrome distinct from non-psychotic major depression, but not a subtype of major depression.

In addition, age was a relatively important variable in the prediction model. To our knowledge, a relationship between age and psychotic symptoms has rarely been reported in patients with major depression; however, several studies have reported the relationship between the age of onset and psychotic symptoms [58]. Our findings are partly supported by a previous report that late-onset elderly depression is associated with more severe depression, psychic anxiety, and gastrointestinal symptoms compared to other age of onset groups [58].

Remarkably, winter and spring as seasons of birth were more important variables than summer and autumn in the prediction model. This is consistent with a previous report of higher seasonality of winter or spring births in patients with schizophrenia compared to the general population [59]. Although the relationship between birth season and depressive symptomatology is controversial [47,48,60], our findings support the trend of a relationship between the season of birth and psychotic symptoms in patients with major depression. An AUC of 0.800–0.900 was considered an acceptable level [61]. Concurrent psychotic symptoms were present only among the Chinese, Japanese, Indian, and Thai patients with depressive disorders. Thus, regional codes (i.e., eastern, southern, or southeastern) and income level codes (i.e., high, upper-middle, or lower-middle income level countries) were included among the variable profiles for the prediction model of the concurrent psychotic symptoms. However, as shown in Figure 3, the region and income level codes were relatively less important variables in the prediction model of psychotic symptoms in patients with depressive disorder. These findings suggest that concurrent psychotic symptoms can be predicted by ethnicity or cultural influences, as well as other sociodemographic and clinical characteristics. Thus, the prediction model of concurrent psychotic symptoms in this study may have been slightly influenced by sampling bias or a factor related to country. In this study, the AUC and overall accuracy of the prediction model were 0.823 and 0.937, respectively, indicating the acceptable predictive performance of the model. However, the model showed low sensitivity and, thus, requires further improvement. As psychotic symptoms have rarely been evaluated, further studies are needed that utilize a strict epidemiological perspective and do not use a convenient sampling method. However, few predictive models exist for concurrent psychotic symptoms in patients with major depression.

Our study had several limitations. First, only 1.4% (*n* = 16) of 1171 patients with depressive disorders presented with concurrent psychotic symptoms. Because of the low rate of concurrent psychotic symptoms, oversampling was required to develop the prediction model [62]. Thus, we applied random oversampling, which involves the random selection of examples from the minority class with replacement and their addition to the training set. Second, the cross-sectional design of the REAP-AD survey made it difficult to elucidate the direction of causation in the prediction model. Third, while our study individually evaluated delusions and hallucinations, the overall psychotic symptoms were evaluated as dichotomous values. Thus, we did not examine predictive models specific to delusions and hallucinations. Further studies are required to develop individual models for specific psychotic symptoms including delusions, hallucinations, disorganized speech, disorganized behaviors, and negative symptoms. Fourth, depressive symptom profiles (i.e., depressed mood, loss of interest and enjoyment, reduced energy and diminished activity, reduced concentration and attention, reduced self-esteem and self-confidence, ideas of guilt and unworthiness, psychomotor agitation or retardation, ideas or acts of self-harm or suicide, disturbed sleep, and diminished appetite) and other symptom profiles (i.e., anxiety symptoms, somatic symptoms) were evaluated as dichotomous variables rather than continuous variables. Therefore, the importance of depressive and other symptom profiles in the prediction model may be affected by their manner of evaluation. Fifth, we cannot rule out the possibility that our findings were affected by a selection bias. As mentioned earlier, the REAP-AD survey used a convenience sampling method instead of a strict epidemiological method, although it is one of the largest international collaborative pharmaco-epidemiological studies. Further studies using structural assessment tools are needed to examine psychotic symptoms, depressive symptom profiles, and other symptom profiles in a strict epidemiological manner. In our study, few patients had psychotic symptoms compared to the whole study sample. This may have affected the prediction model. Despite these limitations, our findings provide a useful basis for further research to establish a predictive model for concurrent psychotic symptoms among Asian patients with depressive disorders.

## 5. Conclusions

The prediction model was clinically applicable for identifying concurrent psychotic symptoms among patients with major depression, whereas we rarely observed concurrent psychotic symptoms. Although the sensitivity requires improvement, the model can be used to detect concurrent psychotic symptoms in patients with major depression. In addition, these findings can serve as a basis for further studies to develop prediction models for concurrent psychotic symptoms in patients with major depression.

## Figures and Tables

**Figure 1 jpm-12-01218-f001:**
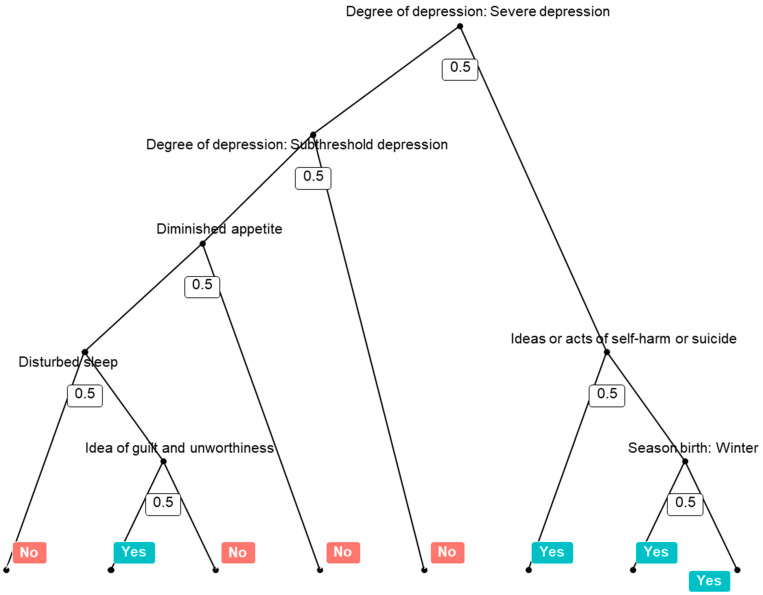
An example of a collection of 500 decision trees in a random forest.

**Figure 2 jpm-12-01218-f002:**
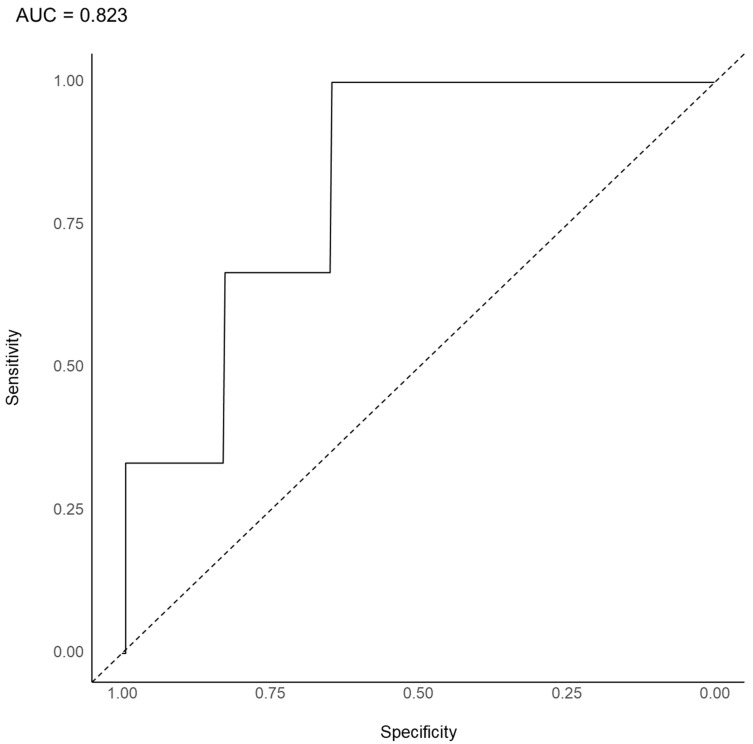
Area under the curve (AUC) of the receiver operating characteristic for predicting psychotic symptoms in patients with depressive disorder (*n* = 1711).

**Figure 3 jpm-12-01218-f003:**
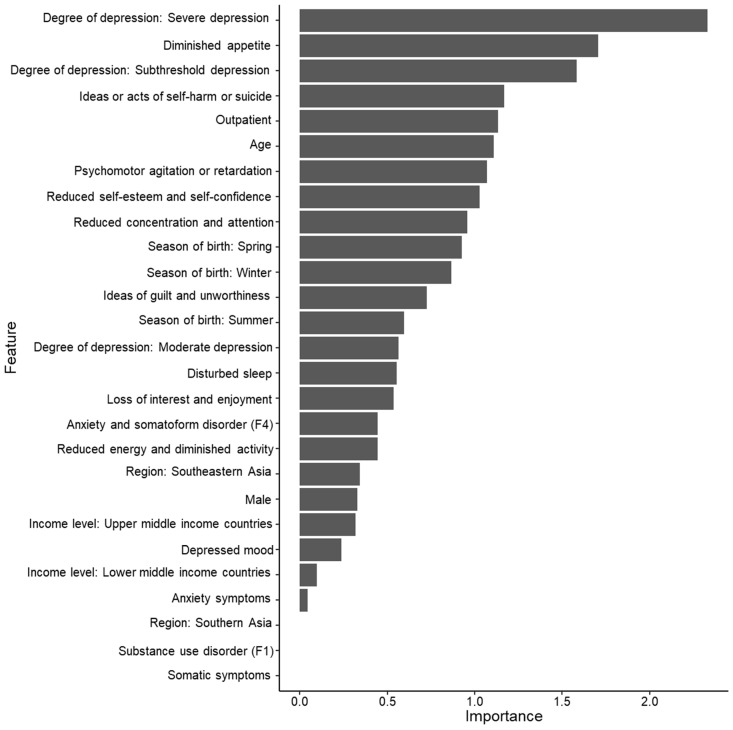
Variable importance in the prediction model of psychotic symptoms in patients with depressive disorder (*n* = 1711).

**Table 1 jpm-12-01218-t001:** Patient characteristics (*n* = 1711).

	Total (*n* = 1171)	Concurrent Psychotic Symptoms		Statistical Coefficient	*p*-Value
		Presence (*n* = 16)	Absence (*n* = 1155)		
Country/SAR				χ^2^ = 22.852	0.007
China, *n* (%)	240 (20.5)	4 (25.0)	236 (20.4)		
Hong Kong, *n* (%)	38 (9.1)	0 (0.0)	28 (9.3)		
Japan, *n* (%)	142 (12.1)	5 (31.3)	137 (11.9)		
Korea, *n* (%)	173 (14.8)	0 (0.0)	183 (15.0)		
Singapore, *n* (%)	38 (3.2)	0 (0.0)	38 (3.3)		
Taiwan, *n* (%)	50 (4.3)	0 (0.0)	50 (3.3)		
India, *n* (%)	130 (11.1)	6 (37.5)	124 (10.7)		
Malaysia, *n* (%)	109 (9.3)	0 (0.0)	109 (9.4)		
Thailand, *n* (%)	144 (12.3)	1 (6.3)	143 (12.4)		
Indonesia, *n* (%)	107 (9.1)	0 (0.0)	107 (9.3)		
Age	48.4 (16.9)	45.3 (17.7)	48.4 (16.9)	t = −0.707	0.490
Male, *n* (%)	477 (40.7)	7 (43.8)	470 (40.7)	χ^2^ = 0.061	0.805
Outpatient, *n* (%)	843 (72.0)	9 (56.3)	834 (72.2)	χ^2^ = 1.993	0.158
Season of birth ^†^				χ^2^ = 5.853	0.119
Spring, *n* (%)	251 (23.6)	1 (10.0)	250 (23.7)		
Summer, *n* (%)	261 (24.6)	4 (40.0)	257 (24.4)		
Autumn, *n* (%)	251 (23.6)	0 (0.0)	251 (23.8)		
Winter, *n* (%)	300 (28.2)	5 (50.0)	295 (28.0)		
Depressive symptom profiles					
Depressed mood, *n* (%)	856 (73.1)	16 (100.0)	840 (98.1)	χ^2^ = 5.969	0.015
Loss of interest and enjoyment, *n* (%)	620 (52.9)	13 (81.3)	607 (52.6)	χ^2^ = 5.216	0.022
Reduced energy and diminished activity, *n* (%)	535 (45.7)	12 (75.0)	523 (45.3)	χ^2^ = 5.617	0.018
Reduced concentration and attention, *n* (%)	347 (29.6)	5 (31.3)	342 (29.6)	χ^2^ = 0.020	0.887
Reduced self-esteem and self-confidence, *n* (%)	268 (22.9)	7 (43.8)	261 (22.6)	χ^2^ = 4.001	0.045
Ideas of guilt and unworthiness, *n* (%)	185 (15.8)	7 (43.8)	178 (15.4)	χ^2^ = 9.527	0.002
Psychomotor agitation or retardation, *n* (%)	266 (22.7)	8 (50.0)	258 (22.3)	χ^2^ = 6.879	0.009
Ideas or acts of self-harm or suicide, *n* (%)	267 (22.8)	6 (37.5)	261 (22.6)	χ^2^ = 1.991	0.158
Disturbed sleep, *n* (%)	747 (63.8)	15 (93.8)	732 (63.4)	χ^2^ = 6.303	0.012
Diminished appetite, *n* (%)	383 (32.7)	11 (68.8)	372 (32.2)	χ^2^ = 9.575	0.002
Degree of depression				χ^2^ = 21.104	<0.0001
Subthreshold, *n* (%)	533 (45.5)	1 (6.3)	532 (46.1)		
Mild, *n* (%)	211 (18.0)	3 (18.8)	208 (18.0)		
Moderate, *n* (%)	296 (25.3)	5 (31.3)	291 (25.2)		
Severe, *n* (%)	131 (11.2)	7 (43.8)	124 (10.7)		
Comorbid symptom profiles					
Anxiety symptoms, *n* (%)	20 (1.7)	0 (0.0)	20 (1.7)	χ^2^ = 0.282	0.595
Somatic symptoms, *n* (%)	15 (1.3)	0 (0.0)	15 (1.3)	χ^2^ = 0.210	0.646
Psychiatric comorbidity					
Anxiety and somatoform disorder (F4), *n* (%)	87 (7.4)	1 (6.3)	86 (7.4)	χ^2^ = 0.033	0.856
Substance use disorder (F1), *n* (%)	20 (1.7)	0 (0.0)	20 (1.7)	χ^2^ = 0.282	0.595

^†^*n* = 1063; SAR, special administrative region.

## Data Availability

Data sharing not applicable.

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
