# Peer review of "A Machine-Learning-Algorithm-Based Prediction Model for Psychotic Symptoms in Patients with Depressive Disorder"

_jpm, 2022, doi:10.3390/jpm12081218_

Round 1

Reviewer 1 Report

Major

1. It seems that the diagnosis was based on ICD-codes (F32, F33) Was psychosis based on diagnostic ICD-codes? Which code? Also, F1, and F4 codes were used for psychiatric comorbidity. More information is needed for the validity of using these codes for each country, as these codes sometimes do not catch the real diagnosis.

 2. Oversampling is needed as psychotic symptoms were rare. How many cases were in the training set, and the test set? There were only 16 cases in total, and I think more discussion is needed on the oversampling technique, and how valid this approach is to a data set with vary rare samples, including its limitations. Also, as these were only from 4 countries, does the country itself serve as a predicitive factor? How does the model change when the variable "country" is inserted with the other variables in the random forest model? It country is an important variable, how can the authors justify that this is not a sampling bias or a factor related to country?

Minor

1. I advise recieving English proofreading.

ex)

Abstract, line 6 : has an advantages -> has advantages

Introduction, 2nd paragraph, line 8 : combination of antidepressant with antipsychotic -> combination of an antidepressant with an antipsychotic

Introduction, 2nd paragraph, last line : with psychotic symptom by -> with psychotic symptoms (?)

Discussion, 5th paragraph 5th line : examinedin -> examined in

Author Response

Major

  1. It seems that the diagnosis was based on ICD-codes (F32, F33) Was psychosis based on diagnostic ICD-codes? Which code? Also, F1, and F4 codes were used for psychiatric comorbidity. More information is needed for the validity of using these codes for each country, as these codes sometimes do not catch the real diagnosis.

We greatly appreciate your kind comments. We completely agree with your comments. We have added the contents in methods as follows:

Under the ICD-10 Classification of Mental and Behavioral Disorders [45], the specifier “with psychotic symptoms” can be coded only in severe depression but not in mild and moderate depression. However, as mentioned earlier, psychotic symptoms can be present in dysthymia, mild to moderate depression, and severe depression. The clinical condition of depression accompanied by concurrent psychotic symptoms is denoted as “psychotic depression.” Thus, irrespective of the presence or absence of concurrent psychotic symptoms, the ICD-10 diagnostic entities were coded exclusively for the severity of depressive episodes or depressive disorder. To increase the validity and reliability of the ICD-10 diagnoses, consensus meetings were held before the initiation of the REAP-AD survey.

  1. Oversampling is needed as psychotic symptoms were rare. How many cases were in the training set, and the test set? There were only 16 cases in total, and I think more discussion is needed on the oversampling technique, and how valid this approach is to a data set with vary rare samples, including its limitations. Also, as these were only from 4 countries, does the country itself serve as a predicitive factor? How does the model change when the variable "country" is inserted with the other variables in the random forest model? It country is an important variable, how can the authors justify that this is not a sampling bias or a factor related to country?

We greatly appreciate your kind comments. We have revised the contents in method, result, and discussion as follows:

In materials and methods:

In addition, using the United Nations classification, the 10 Asian countries and SARs were geographically divided into eastern Asia (mainland China, Hong Kong, Japan, Korea, and Taiwan), southern Asia (India), and southeast Asia (Indonesia, Malaysia, Singapore, and Thailand). Furthermore, using the World Bank list of economies, the countries and SARs were economically categorized as high-income (Hong Kong, Japan, Korea, Singapore, and Taiwan), upper-middle (China, Malaysia, and Thailand), and lower-middle income (India and Indonesia) and SARs [42-44].

Variable Profiles for the Prediction Model of Concurrent Psychotic Symptoms

This study used sociodemographic data (i.e., region [Eastern Asia, Southern Asia, or Southeastern Asia], income level [i.e., high, upper-middle, or lower-middle income level countries], age, sex, or season of birth [i.e., spring, summer, autumn, or winter]), the ten individual depressive symptom profiles, degree of depression (i.e., subthreshold, mild, moderate, or severe), comorbid anxiety or somatic symptoms, psychiatric comorbidity (i.e., substance use disorder [F1] or anxiety and somatoform disorder [F4]), and outpatient or inpatient admission as variables for the prediction model of concurrent psychotic symptoms.

The dataset was divided into training and test sets at a 7:3 ratio among the 1,171 patients with depressive disorders.

In results;

Prediction Model Performance for Concurrent Psychotic Symptoms and its Variable Importance

A collection of 500 decision trees, which were generated in this analysis, was used by a random forest. An example of 500 decision trees was presented, as shown in Fig. 1. The AUC under the ROC curve was 0.823 (Fig. 2). The overall accuracy was 0.931 (95% confidence interval, 0.897–0.956), the sensitivity was 0.047, the specificity was 0.993, the positive predictive value was, 0.333; and negative predictive value was 0.937. Severe levels of depression (degree of depression) were the most important variables in the prediction model for concurrent psychotic symptoms, followed by diminished appetite, subthreshold (degree of depression), ideas or acts of self-harm or suicide, outpatient status, age, psychomotor retardation or agitation, and others (Fig. 3).

In discussion:

Concurrent psychotic symptoms were present only among the Chinese, Japanese, Indian, and Thai patients with depressive disorders. Thus, regional codes (i.e., Eastern, Southern, or Southeastern) and income level codes (i.e., high, upper-middle, or lower-middle income level countries) were included among the variable profiles for the prediction model of the concurrent psychotic symptoms. However, as shown in Fig. 3, the region and income level codes were relatively less important variables in the prediction model of psychotic symptoms in patients with depressive disorder. These findings suggest that concurrent psychotic symptoms can be predicted by ethnicity or cultural influences, as well as other sociodemographic and clinical characteristics. Thus, the prediction model of concurrent psychotic symptoms in this study may have been slightly influenced by sampling bias or a factor related to country

Minor

  1. I advise recieving English proofreading.

ex)

Abstract, line 6 : has an advantages -> has advantages

Introduction, 2nd paragraph, line 8 : combination of antidepressant with antipsychotic -> combination of an antidepressant with an antipsychotic

Introduction, 2nd paragraph, last line : with psychotic symptom by -> with psychotic symptoms (?)

Discussion, 5th paragraph 5th line : examinedin -> examined in

We greatly appreciate your kind comments. We have thoroughly revised the English expression. We have additionally added English proofreading.

Reviewer 2 Report

Conclusively, I suggest accepting this article after a minor revision. I have no opinions on all the discussions in this manuscript; however, the authors may need to make the following two revisions:

  1.  The authors obtained an AUC equal to 0.835 with the proposed model, but I suggest adding the results provided by a baseline model such as a decision tree to demonstrate that this AUC value is sufficiently good.
  2. The authors created Figure 2 to demonstrate the variable importance. For example, the two most important variables are the degree of depression and diminished appetite. However, I think readers of this manuscript would like to visualize the classification results with these two variables, not just a list ranking the importance of each variable.

Author Response

Conclusively, I suggest accepting this article after a minor revision. I have no opinions on all the discussions in this manuscript; however, the authors may need to make the following two revisions:

  1.  The authors obtained an AUC equal to 0.835 with the proposed model, but I suggest adding the results provided by a baseline model such as a decision tree to demonstrate that this AUC value is sufficiently good.

We greatly appreciate your kind comments. We have added fig. 1 as follows: Fig. 1. An example of a collection of 500 decision trees in a random forest

  1. The authors created Figure 2 to demonstrate the variable importance. For example, the two most important variables are the degree of depression and diminished appetite. However, I think readers of this manuscript would like to visualize the classification results with these two variables, not just a list ranking the importance of each variable.

 We greatly appreciate your kind comments. We have revised fig. 2.

Conclusively, I suggest accepting this article after a minor revision. I have no opinions on all the discussions in this manuscript; however, the authors may need to make the following two revisions:

  1.  The authors obtained an AUC equal to 0.835 with the proposed model, but I suggest adding the results provided by a baseline model such as a decision tree to demonstrate that this AUC value is sufficiently good.

We greatly appreciate your kind comments. We have added fig. 1 as follows: Fig. 1. An example of a collection of 500 decision trees in a random forest

  1. The authors created Figure 2 to demonstrate the variable importance. For example, the two most important variables are the degree of depression and diminished appetite. However, I think readers of this manuscript would like to visualize the classification results with these two variables, not just a list ranking the importance of each variable.

 We greatly appreciate your kind comments. We have revised fig. 2.

Round 2

Reviewer 1 Report

The authors have answered all my questions sufficiently. I have no other comments

Author Response

According to the reviewer comment, we have revised fig. 2.
